# Analysis of *Puumala orthohantavirus* Genome Variants Identified in the Territories of Volga Federal District

**DOI:** 10.3390/tropicalmed7030046

**Published:** 2022-03-06

**Authors:** Emmanuel Kabwe, Walaa Al Sheikh, Anton F. Shamsutdinov, Ruzilya K. Ismagilova, Ekaterina V. Martynova, Olesia V. Ohlopkova, Yuri A. Yurchenko, Tatiana A. Savitskaya, Guzel S. Isaeva, Svetlana F. Khaiboullina, Albert A. Rizvanov, Sergey P. Morzunov, Yuriy N. Davidyuk

**Affiliations:** 1OpenLab “Gene and Cell Technologies ”, Institute of Fundamental Medicine and Biology, Kazan Federal University, 420008 Kazan, Russia; loloa123355@gmail.com (W.A.S.); shamsutdinov2006@yandex.com (A.F.S.); ignietferro.venivedivici@gmail.com (E.V.M.); sv.khaiboullina@gmail.com (S.F.K.); rizvanov@gmail.com (A.A.R.); smorzunov@med.unr.edu (S.P.M.); 2Kazan Research Institute of Epidemiology and Microbiology, 420012 Kazan, Russia; tatasav777@mail.ru (T.A.S.); guisaeva@rambler.ru (G.S.I.); 3OpenLab “Omics Technology”, Institute of Fundamental Medicine and Biology, Kazan Federal University, 420008 Kazan, Russia; ruz-ismagilova@yandex.ru; 4State Research Center of Virology and Biotechnology “Vector”, Rospotrebnadzor, World-Class Genomic Research Center for Biological Safety and Technological Independence, Federal Scientific and Technical Program on the Development of Genetic Technologies, 630559 Koltsovo, Russia; ohlopkova_ov@vector.nsc.ru; 5Hygienic and Epidemiological Center for Novosibirsk Region, 630099 Novosibirsk, Russia; yurons@ngs.ru; 6Institute of Systematics and Ecology of Animals, Siberian Branch of the Russian Academy of Sciences, 630091 Novosibirsk, Russia; 7Department of Pathology, University of Nevada, Reno, NV 89557, USA

**Keywords:** *Puumala orthohantavirus* genome, genetic variability, phylogenetic analysis, reassortation

## Abstract

Hemorrhagic fever with renal syndrome (HFRS) is a zoonotic disease commonly diagnosed in the Volga Federal District (VFD). HFRS is caused by *Puumala* *orthohantavirus* (PUUV), and this virus is usually detected in bank voles as its natural host (*Myodes glareolus*). The PUUV genome is composed of the single-stranded, negative-sense RNA containing three segments. The goal of the current study is to identify genome variants of PUUV strains circulating in bank voles captured in the Udmurt Republic (UR) and Ulyanovsk region (ULR). The comparative and phylogenetic analysis of PUUV strains revealed that strains from Varaksino site UR are closely related to strains previously identified in the Pre-Kama area of the Republic of Tatarstan (RT), whilst strains from Kurlan and Mullovka sites ULR are similar to strains from the Trans-Kama area of the RT. It was also found that Barysh ULR strains form a separate distinct group phylogenetically equidistant from Varaksino and Kurlan–Mullovka groups. The identified groups of strains can be considered as separate sub-lineages in the PUUV Russian genetic lineage. In addition, the genomes of the strains from the UR, most likely, were formed as a result of reassortment.

## 1. Introduction

The Volga Federal District (VFD) belong to the Russian region where many cases of hemorrhagic fever with renal syndrome (HFRS) are registered annually, with *Puumala*
*orthohantavirus* (PUUV) identified as the main cause of the illness [1,2]. Among the territories of the VFD, the most affected regions with HFRS are the Republic of Tatarstan (RT), the Republic of Bashkortostan, the Udmurt Republic (UR), the Republic of Mordovia, and the Penza region [3]. HFRS in the VFD presents a mild to moderate clinical form of symptoms, often referred to as nephropatia epidemica (NE) in Europe [1,4,5,6,7]. In bank voles (*Myodes glareolus*), the natural host of PUUV, the virus is persistent and causes a lifelong asymptomatic infection [8]. PUUV circulation in the environment is attributed to the continuous reinfection of the natural hosts via horizontal transmission [9]. Human disease occurs via inhaling virus-contaminated aerosols excreted by the host through urine, feces, and saliva or through direct contact with an infected rodent [4,10]. The illness has an acute onset with flu-like symptoms, and it progresses with the development of impaired renal function and disturbed hemodynamic symptoms [4,8]. Usually, the mortality rate of NE is less than 1% [10]. Whether the high number of PUUV infections in VFD regions can be linked to the specific viral genotype remain unknown [11,12,13]. Currently, limited data are available addressing the relationship between increased incidences of HFRS, clinical manifestations of the disease, and PUUV genotypes in the territories of the VFD.

PUUV is a tri-segmented negative polarity RNA virus belonging to the genus *Orthohantavirus*, family *Hantaviridae* [14]. The genome consists of three segments, the small (S) 1826–1830 nucleotides (nt), the medium (M) 3682 nt, and the large (L) 6530–6562 nt. The segments encode for a nucleocapsid (N) protein of 433 amino acids (aa), glycoprotein precursor Gn and Gc (GPC) of 1148 aa, and the RNA-dependent RNA polymerase (RdRp) of 2156 aa [12,15,16].

The PUUV genome is highly variable, where variations correlate with geographical location of the bank voles in the regions of Europe [17,18,19]. Currently, eight genetic lineages of PUUV have been discovered in their natural reservoirs: (i) a Central European (CE) lineage (found in France, Belgium, Germany, and Slovakia); (ii) an Alpe-Adrian (ALAD) lineage (present in Austria, Slovenia, Croatia, and Hungary); the third (iii) lineage is Danish (DAN) and includes strains from Denmark; (iv) a South Scandinavian (S-SCA) lineage consist of strains from Norway and southern Sweden; (v) a North Scandinavian (N-SCA) lineage is found in northern Sweden; (vi) a Finnish (FIN) lineage circulating in Finland, Russian Karelia, and western Siberia; (vii) a Russian (RUS) lineage with strains found in Russia and Estonia; and the eighth (viii) is a Latvian (LAT) lineage with strains from Latvia, including strains from northeast Poland [15,20].

In our previous works, the PUUV strains’ genetic variability in the Pre-Kama and Trans-Kama areas of the RT were investigated [21,22]. Nevertheless, little is known about the genetic diversity of the PUUV strains in the other VFD territories. Therefore, the goal of the current study was to investigate the genetic characteristics of the PUUV genome discovered in bank voles from the UR and Ulyanovsk region (ULR). In addition, we determined the phylogenetic relationship with PUUV strains circulating in other territories of the VFD.

## 2. Materials and Methods

Frozen rodent lung tissue samples and information about trapping localities were obtained from Federal Healthcare Institute “Centre for Hygiene and Epidemiology in the Republic of Tatarstan (Tatarstan)”. The Institutional Review Board of the Kazan Federal University approved this study, and the above institution presented rodent tissues according to the guidelines approved under the corresponding protocol. 

For initial screening of the samples for the presence of PUUV RNA Syntol reagents “OM-Screen-HFRS-RT” Kit (SYNTOL, Moscow, Russia) was used in line with the manufacturer’s instructions. Furthermore, total RNA was extracted from the PUUV RNA positive lung tissues of bank voles using TRIzol Reagent (Invitrogen Life Technologies^TM^, Waltham, MA, USA), according to the manufacturer’s instructions. Reverse transcription for cDNA synthesis was performed using Thermo Scientific RevertAid Reverse Transcriptase (“Thermo Fisher Scientific”, Waltham, MA, USA) following the manufacturer’s recommendations. PCR amplification was undertaken using Taq-polymerase (“Evrogen”, Moscow, Russia) as specified by the manufacturer. Forward and reverse primers used for PCRs amplification and sequencing analysis are published by Davidyuk et al. [21].

PCR amplicons were purified using Isolate II PCR and Gel Kit (“Bioline”, London, UK) and automated sequenced using ABI PRISM 310 big Dye Terminator 3.1 sequencing kit (ABI, Waltham, MA, USA) per specifications of the manufacturer. Sequences were deposited in the GenBank database under the accession no. OL840976-OL840981 and OL840983-840987 for complete CDS S segment, OL860965-OL860970 and OL840988-OL840993 for partial M segment, and OL840617-OL840627 for partial L segment. 

Multiple nucleotide sequence alignments of the PUUV strains were performed using MegAlign program (Clustal W algorithm) located in the DNASTAR software package Lasergene (DNASTAR, Madison, WI, USA; https://www.dnastar.com/, accessed on 12 January 2022) and MEGA v6.0 [23]. Phylogenetic trees were inferred using Maximum Likelihood (ML) method incorporated in Mega v6.0. [23]. The Tamura–Nei model for all the three segments was used. The bootstrap values less than 70% are not shown on the trees. For the phylogeny analysis, reference sequences of PUUV strains from multiple lineages were obtained from GenBank, including those previously characterized from the RT and other regions of Russia (Table 1). Sequences of *Tula orthohantavirus* were used as outgroups for the phylogenetic analysis of the S segment (AF164093), M segment (NC_005228) and L segment (NC_005226). 

## 3. Results and Discussion

In total, 160 rodent samples were collected in the VFD region in 2019: 100 samples from ULR and 60 from UR (Figure 1). 

In total, 43 bank vole lung tissues from VFD were found to be positive for PUUV RNA after the initial screening. The complete coding sequences (CDS) (1302 bp) for the S, partial M (1014 bp), and L (603 or 665 bp) segment sequences were only obtained from 11 samples and used for analysis. All PUUV strains obtained were named to include their corresponding virus acronym name, trapping region, host acronym name/number, and trapping year (for example, for PUUV/Varaksino/MG_1937/2019, we will use MG1937 in short).

The comparison analysis of the CDS S segment revealed high similarities among the sequences obtained from PUUV strains captured within the same sites (Table 2). The S segment nucleotide sequences formed three groups corresponding to geographical location of their bank vole trapping sites: the Varaksino group (six strains), the Kurlan and Mullovka (three strains) group, and the Barysh group (two strains). The Varaksino group contained PUUV strains from UR (Varaksino site) with sequence identities equal to 100%. The identity among strains in the Kurlan and Mullovka group from ULR (Kurlan and Mullovka sites will furthermore be referred to as “Kurlan–Mullovka”) was 98.5–99.5%, whilst the identity of the Barysh group strains also from ULR (Barysh site) was 99.8% (Table 2). Analysis of the investigated sequences among the groups (Varaksino, Kurlan–Mullovka, and Barysh groups) showed a lower identity. The identity between strains from the Varaksino and Kurlan–Mullovka groups were in the range of 94.2–95.1%, whilst the Varaksino and Barysh group strains were in the range of 93.8% to 93.9%. Interestingly, in the Kurlan–Mullovka and Barysh group strains obtained from ULR, the identities between them were 93.2% to 93.7 %. This value of identity is observed among the PUUV strains, which are geographically distantly located from each other. The percentage identity among all the strains from the three groups and other strains belonging to the RUS genetic lineage was in the range of 92.9% to 98.5% (Table 3). The sequence identities between all RUS-lineage PUUV strains and all other genetic lineages was 81.3% to 86.5%. Therefore, we can conclude that all the obtained sequences belong to the RUS genetic lineage.

The N protein amino acid sequences (aa) of the investigated PUUV strains had 99.8–100% identity among the strains within each group. These sequences were 98.8–100.0% similar when compared among the groups and strains previously identified in RT and other strains of the RUS lineage. The lower aa sequence identity was observed when comparing identified strains with strains of FIN, CE, and N-SCA lineages (95.8–97.5%). Furthermore, we found four aa substitutions in the most variable region of the N-protein. The E238D mutation was found only in strains from the Varaksino group and previously identified strains Kazan and PUUV/Naberezhnye Chelny/MG_260/2015; V260I was detected in strains from the Varaksino group and is found in strains previously detected in Pre-Kama area [21], while isolates from the Kurlan–Mullovka group have K242R together with strains from RT: It should be noted that most of the samples from the Pre-Kama area have K aa substitution, whilst strains circulating in the Trans-Kama area have R. Interestingly, the E250D aa substitution was only found in strains from the Barysh group.

Similar to that for S segments, the M segment sequences formed three groups corresponding to the geographic localization of the trapping sites; the Varaksino group contains strains from UR, whilst the Kurlan–Mullovka and Barysh groups include isolates from ULR. The identity among sequences within each trapping site was in the range of 99.0–100.0%, while the similarity among samples from the Varaksino, Kurlan–Mullovka, and Barysh groups was 91.6% to 94.5% (Table 2). Similar to that in the S segment sequences, the lower identity was shown when comparing sequences from the Barysh group and other groups (91.6–93.0%). When comparing all the investigated strains from the 3 groups to sequences from the RUS genetic lineage, the identity ranged from 91.5% to 96.0% with Samara and Kazan strains, whilst that with the CG1820 and DTK/Ufa strains was 85.3% to 87.5%. The similarity among the investigated strains and PUUV strains from FIN, CE, and N-SCA lineages was 79.3% to 84.1% (Table 3).

Analysis of the GPC aa sequences showed that strains from ULR and UR were 99% identity within each group. In addition, these aa sequences were 98.1% identity when compared among the groups. The identity between ULR and UR aa sequences and previously identified PUUV strains Samara, Kazan, CG1820, and Ufa, which belong to the RUS lineage, was 98.2% to 99.7%. The lower identity was observed when the investigated aa sequences were compared to PUUV isolates from Central and Northern Europe (FIN, CE, and N-SCA lineages), with identity of 91% to 97% (Table 3). The analysis of these aa sequences identified several aa substitutions in strains from the Varaksino and Kurlan–Mullovka groups such as the Q → L/R/P at position 608. 

The nucleotide sequence comparison of the partial L segment of the PUUV strains from ULR and UR showed low nucleotide sequence identity among the samples trapped at the same site. Similar to the S and M segments, the L segment sequences also formed 3 groups (Varaksino, Kurlan–Mullovka, and Barysh) and revealed a similar nucleotide identity level with the range of 98.0–100.0% within each group. The sequence identity between the strains from the Varaksino, Kurlan–Mullovka, and Barysh groups varied from 86.2% to 93.7%. Likewise, the sequences from the Barysh group had 84.9–86.2% nucleotide identity when compared to the Varaksino and Kurlan–Mullovka groups’ sequences (Table 2). Though with lower identity, these strains belong to the RUS genetic lineage. The Varaksino and Kurlan–Mullovka groups’ sequences demonstrated 91.7–95.5% nucleotide similarities compared to our previous identified RT strains, Kazan, and Samara strains. In addition, they shared 85.4–86.1% nucleotide sequence identity compared to CG1820 and Ufa strains, whilst the Barysh group sequence’s identity ranged from 84.1% to 86.5% with strains belonging to the RUS lineage. All 11 investigated sequences displayed lower identity with the PUUV sequences of the FIN, CE, and N-SCA genetic lineages, with the level of similarities varying from 78.8% to 83.3% (Table 3).

These data suggest that the L segment nucleotide sequences of the PUUV strains obtained in the current study are characterized by high variability compared to the sequences of the S and M segment. Despite the low nucleotide sequence identity, the identity of the RdRp aa sequences among all studied strains exceeded 98.5%. In addition, the identity of these strains when compared to reference strains of RUS lineage was 98.2%. The identity of aa sequences of the identified strains when compared to the strains of FIN, CE, and N-SCA lineages was 94.1%. Therefore, we postulate that the sequences of the L segment belong to the RUS genetic lineage circulating in the VFD.

Further analysis of RdRp aa substitutions revealed that the Barysh group strains carried a specific mutation S1114A, which was not detected in other PUUV strains or the previous strains identified in Russia. In addition to this aa, the I1070L substitution was found similar to that observed in the Ufa strain.

The inferred phylogenetic trees based on the complete CDS S (1302 nt, position 43–1344 on Kazan strain), as well as partial M (1014 nt, position 1499–2512 on Kazan strain) and L (603 nt, position 958–1560 on Kazan strain) segments, have similar topology (Figure 2, Figure 3 and Figure 4, respectively). The ULR and UR PUUV sequences formed three subclades, Varaksino, Kurlan–Mullovka, and Barysh, corresponding to geographical location of the bank vole trapping sites in the ULR and UR area of the VFD. 

The location of the Kurlan–Mullovka and Barysh subclades were identical on all three phylogenetic trees. On all trees, a well-supported Kurlan–Mullovka subclade is clustered with previously identified strains from the Western Trans-Kama area of the RT [22], whilst the Barysh group strains PUUV/Barysh/MG_1769/2019 and PUUV/Barysh/MG_1770/2019 formed an independent subclade with good support. At the same time, the location of subclade Varaksino on the phylogenetic trees differed. On the S and M segment trees, this subclade is branched with strains from Pre-Kama area of the RT (Figure 2 and Figure 3). Furthermore, this subclade on phylogenetic tree for the L segment is more related to the strains from Trans-Kama area of the RT than to the Pre-Kama area strains (Figure 4). 

The closer phylogenetic relationship of PUUV strains from Varaksino and the Western Trans-Kama region were somewhat unexpected. The inconsistencies in the location of subclade Varaksino on the phylogenies suggest that the PUUV genome could be chimeric, comprising parts with different origins. In addition, these observations could be the result of either whole intra-genome segment reassortment or RNA recombination. A similar phenomenon was previously reported among the PUUV strains from the N-SCA and FIN genetic lineages [17]. In Finland, researchers also found some PUUV genomes that consisted of reassortants of the segments, which came from different genetic variants of the FIN genetic lineage [9]. One could suggest with confidence that the genome of the investigated strains contain the S, M, and L segments that originated from the genetically distinct PUUV strains belonging to different variants of RUS genetic lineage, consequently supporting the recombination or reassortment hypothesis. Furthermore, a comparative analysis of PUUV nucleotide sequences from Varaksino revealed three specific mutations at the 1209G, 1266G, and 1468T positions. These mutations were also specific for strains from the Pre-Kama area. However, no specific nucleotide sequence mutations were detected between Varaksino strains and strains circulating in the Trans-Kama area. Thus, it can be assumed that Varaksino strains are more related to strains from the Pre-Kama area, despite the different location of Varaksino subclade on the phylogenetic tree of the L segment. In addition, it appears that migrations of bank voles caring ancestral Varaksino strains to UR could not have happened from the south as proposed earlier because the Kama River acts as a barrier [24] but from the east [25].

The observation of a genetically distinct of PUUV group in ULR could suggest the existence of another PUUV genetic sub-lineage in the central areas of ULR. This sub-lineage could have been formed due to the isolation of bank vole populations because of the Volga River, which act as a natural barrier to the bank vole migrations.

Based on the phylogenetic analysis of PUUV S, M, and L segment sequences obtained from the bank voles captured in the ULR and UR of the VFD, we propose the existence of three genetically distinct sub-lineages of PUUV strains within the RUS genetic lineage (Sub-lineages I, II, and III). Strains in sub-lineage I are commonly found in the Pre-Kama and Trans-Kama areas of the RT [21,22], including parts of the ULR located on the left bank of Volga River and Samara region. These strains demonstrate high genetic identity of nucleotide and aa sequences along the territories of the left bank of the Volga River. Two PUUV strains identified in the Barysh located in ULR on the right bank of the Volga River did not cluster with any previously identified strains. Partial L segment nucleotide sequence identity among Barysh strains and sub-lineage I strains ranges from 84.7% to 86.4% (Table 2 and Table 3). Barysh strains’ aa sequences contain two aa substitutions when compared to the Sub-lineage I strains. Therefore, these strains could form a genetically distinct sub-lineage II, which is phylogenetically separated from sub-lineage I. However, there is limited data on the PUUV strains circulating on the right bank of the Volga River to make an unambiguous conclusion. Additionally, it should be noted that CG1820 and Ufa strains from Bashkiria used for comparison and phylogenetic analysis could be the representatives of the third sub-lineage in the RUS genetic lineage. Partial M and L segment nucleotide sequences identities of CG1820 and Ufa strains, when compared with strains from other regions of the VFD, do not exceed 87.5% and 86.1%, respectively (Table 3). Corresponding aa sequences of Bashkiria strains differ from other VFD strains by 5–10 and 2–3 aa substitutions for GPC and RdRp, respectively. However, more additional evidence, including investigating the PUUV genome variants circulating in Bashkiria, is required to test this assumption.

## 4. Conclusions

Based on the analysis of the PUUV complete CDS S, partial M, and L segment sequences, several variants of PUUV genome were identified in the bank vole populations as circulating in the VFD and demonstrated high level of variabilities. The obtained results can be useful for unraveling the relationships between the PUUV variants circulating in different regions of the VFD. Comparative and phylogenetic analysis suggested that strains from Varaksino site in UR, Kurlan and Mullovka sites, and the Barysh site in the ULR belong to two different PUUV sub-lineages within the RUS genetic lineage.

## Figures and Tables

**Figure 1 tropicalmed-07-00046-f001:**
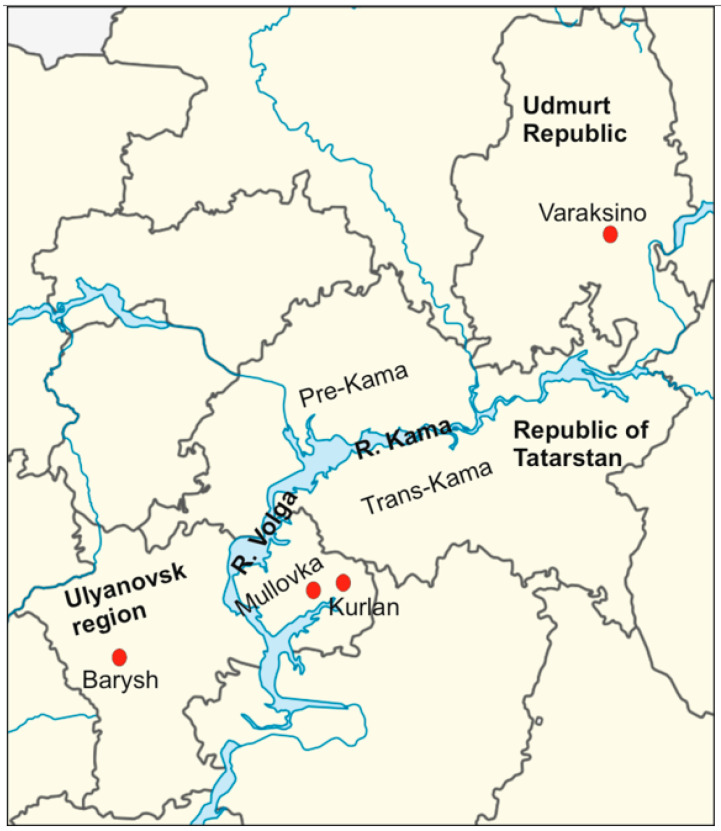
Geographical location of rodent trapping sites in the ULR and UR of the VFD. The red dots represent the trapping sites. The map was modified from free blank SVG vector map of Russia (MAPSVG, from https://mapsvg.com/maps/russia, accessed on 12 January 2022).

**Figure 2 tropicalmed-07-00046-f002:**
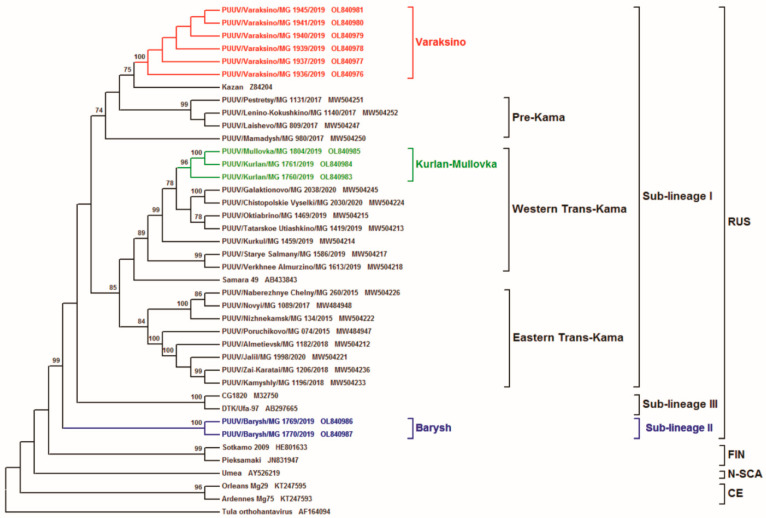
Phylogenetic tree calculated based on the complete coding region (CDS) (1302 bp) of PUUV S segment sequences obtained from bank voles trapped in ULR and UR. The sequence position was aligned against GenBank PUUUV Kazan strain, accession number Z84204.

**Figure 3 tropicalmed-07-00046-f003:**
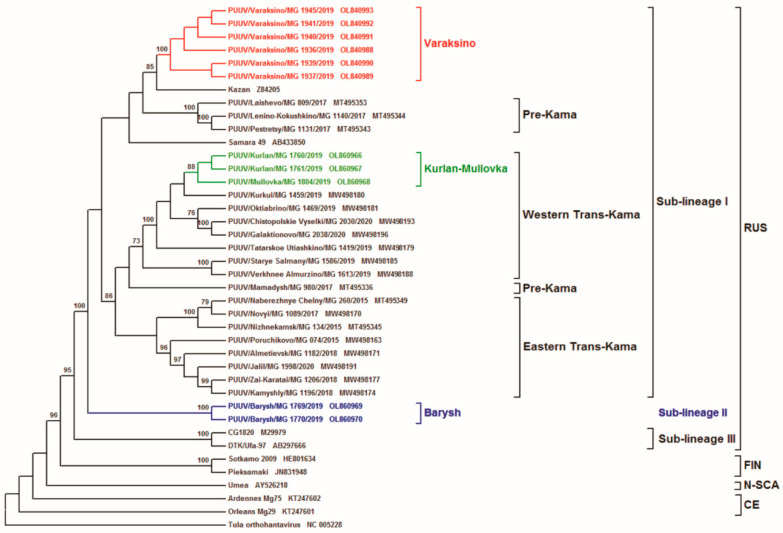
Phylogenetic tree calculated based on the partial (1014 bp) of PUUV M segment sequences obtained from bank voles trapped in ULR and UR. The sequence position was aligned against GenBank Kazan, accession number Z84205.

**Figure 4 tropicalmed-07-00046-f004:**
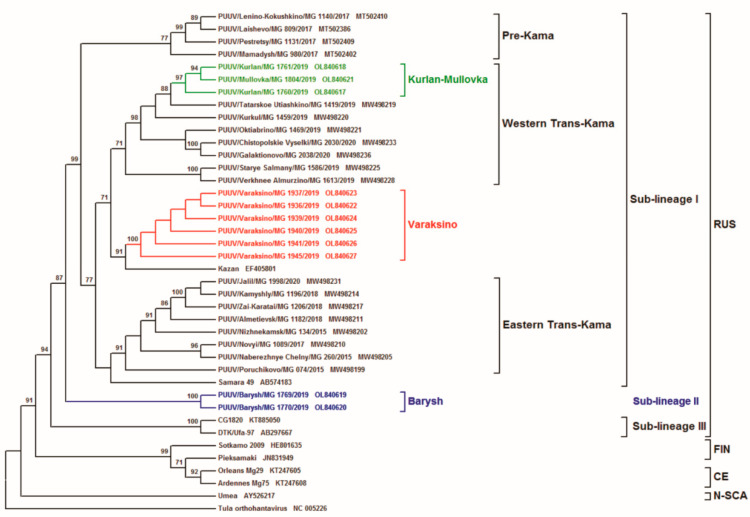
Phylogenetic tree calculated based on the partial (603 bp) of PUUV L segment sequences obtained from bank voles trapped in ULR and UR areas. The sequence position was aligned against GenBank PUUV Kazan strain, accession number EF405801.

**Table 1 tropicalmed-07-00046-t001:** The list of PUUV strains and *Tula orthohantavirus* obtained from GenBank database and used for comparison analysis in this study.

Strain	Location in the RT	Abbreviation	GenBank Accession Number		
			S Segment	M Segment	L Segment
PUUV/Poruchikovo/MG_074/2015	Eastern Trans-Kama	MG074	MW484947	MW498163	MW498199
PUUV/Nizhnekamsk/MG_134/2015	Eastern Trans-Kama	MG134	MW504222	MT495345	MW498202
PUUV/Naberezhnye Chelny/MG_260/2015	Eastern Trans-Kama	MG260	MW504226	MT495349	MW498205
PUUV/Laishevo/MG_809/2017	Pre-Kama	MG809	MW504247	MT495353	MT502386
PUUV/Mamadysh/MG_980/2017	Pre-Kama	MG980	MW504250	MT495336	MT502402
PUUV/Novyi/MG_1089/2017	Eastern Trans-Kama	MG1089	MW484948	MW498170	MW498210
PUUV/Pestretsy/MG_1131/2017	Pre-Kama	MG1131	MW504251	MT495343	MT502409
PUUV/Lenino-Kokushkino/MG_1140/2017	Pre-Kama	MG1140	MW504252	MT495344	MT502410
PUUV/Almetievsk/MG_1182/2018	Eastern Trans-Kama	MG1182	MW504212	MW498171	MW498211
PUUV/Zai-Karatai/MG_1206/2018	Eastern Trans-Kama	MG1206	MW504236	MW498177	MW498217
PUUV/Kamyshly/MG_1196/2018/2018	Eastern Trans-Kama	MG1196	MW504233	MW498174	MW498214
PUUV/Tatarskoe Utiashkino/MG_1419/2019	Western Trans-Kama	MG1419	MW504213	MW498179	MW498219
PUUV/Kurkul/MG_1459/2019	Western Trans-Kama	MG1459	MW504214	MW498180	MW498220
PUUV/Oktiabrino/MG_1469/2019	Western Trans-Kama	MG1469	MW504215	MW498181	MW498221
PUUV/Starye Salmany/MG_1586/2019	Western Trans-Kama	MG1586	MW504217	MW498185	MW498225
PUUV/Dzhalil/MG_1998/2020	Eastern Trans-Kama	MG1998	MW504221	MW498191	MW498231
PUUV/Verkhnee Almurzino/MG_1613/2019	Western Trans-Kama	MG1613	MW504218	MW498188	MW498228
PUUV/Chistopolskie Vyselki/MG_2030/2020	Western Trans-Kama	MG2030	MW504224	MW498193	MW498233
PUUV/Galaktionovo/MG_2038/2020	Western Trans-Kama	MG2038	MW504245	MW498196	MW498236
Puu/Kazan		«Kazan»	Z84204	Z84205	EF405801
Samara_49/CG/2005		«Samara»	AB433843	AB433850	AB574183
CG1820		CG1820	M32750	M29979	KT885050
DTK/Ufa-97		«Ufa»	AB297665	AB297666	AB297667
Sotkamo 2009		«Sotkamo»	HE801633	HE801634	HE801635
PUUV/Pieksamaki/human_lung/2008		«Pieksamaki»	JN831947	JN831948	JN831949
Umea/hu		«Umea»	AY526219	AY526218	AY526217
PUUV/Orleans/Mg29/2010		«Orleans»	KT247595	KT247601	KT247605
PUUV/Ardennes/Mg75/2011		«Ardennes»	KT247593	KT247602	KT247608
*Tula orthohantavirus*		«Tula»	AF164094	NC_005228	NC_005226

**Table 2 tropicalmed-07-00046-t002:** Nucleotide sequence identity of the PUUV complete CDS S, partial M, and L segments recovered from bank voles (%) captured in ULR and UR areas.

PUUV Strain	Identity, %								
	S Segment			M Segment			L Segment		
	Udmurt Republic	Ulyanovsk Region		Udmurt Republic	Ulyanovsk Region		Udmurt Republic	Ulyanovsk Region	
	Varaksino	Kurlan–Mullovka	Barysh	Varaksino	Kurlan–Mullovka	Barysh	Varaksino	Kurlan–Mullovka	Barysh
Varaksino	100.0	94.2–95.1	93.8–93.9	99.9–100.0	93.7–94.5	92.9–93.0	100.0	93.0–93.7	85.9
Kurlan–Mullovka		98.5–99.5	93.2–93.7		99.0–99.7	91.6–92.1		98.0–99.3	86.2–86.4
Barysh			99.8			100.0			100.0

**Table 3 tropicalmed-07-00046-t003:** Percentage identities between ULR and UR PUUV complete CDS segment S, partial M, and L sequences identified in different trapping sites and belonging to RUS, FIN, CE, and N-SCA genetic lineages.

Location	RUS					FIN		CE		N-SCA
	RT	Samara	Kazan	CG1820	Ufa	Sotkamo	Pieksamaki	Ardennes	Orleans	Umea
S segment										
Varaksino	94.1–96.2	94.9	96.5	94.3	94.4	84.9	86.2	83.2	83.3	83.6
Kurlan–Mullovka	94.2–98.5	96.1–96.4	94.5–94.9	93.8–94.3	93.9–94.4	85.3–85.5	86.3–86.6	83.3–84.1	83.0–83.2	84.6–84.8
Barysh	92.9–94.2	93.9–94.0	93.5	93.6–93.8	93.7–93.9	85.7–85.8	85.7	83.6	82.6	84.6
M segment										
Varaksino	93.0–95.4	93.7–93.8	95.9–96.0	86.8–86.9	87.4–87.5	83.6–83.7	83.1–83.2	81.1–81.2	79.3–79.4	81.2–81.3
Kurlan–Mullovka	93.0–98.8	92.4–92.8	93.8–94.6	85.3	85.9	84.1–84.5	83.9–84.1	81.1–81.5	80.3–80.5	81.2–81.5
Barysh	90.6–93.3	91.5	91.9	86.0	86.6	83.2	82.5	81.3	80.2	81.0
L segment										
Varaksino	92.7–95.0	93.7	95.5	85.9	85.9	83.1	83.7	81.6	82.8	79.9
Kurlan–Mullovka	91.7–97.7	93.0–93.4	92.7–93.7	85.4–86.1	85.4–86.1	82.3–82.6	82.9–83.1	81.8	82.6–82.9	79.6–79.9
Barysh	84.7–87.2	85.7	85.6	84.1	84.1	82.1	80.6	78.9	79.9	80.3

## Data Availability

Not applicable.

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
