# Peer review of "Analysis of Puumala orthohantavirus Genome Variants Identified in the Territories of Volga Federal District"

_tropicalmed, 2022, doi:10.3390/tropicalmed7030046_

Round 1

Reviewer 1 Report

Thank your for this article. All my comments are in the pdf enclosed.

Reviewer 2 Report

The overall assessment of the work is high. The only remark concerns the title of the article. The mention of the Novosibirsk region is incongruously, because studies based on materials from this region are not presented in the article. In our opinion, correct title will be "Analysis of Puumala Orthohantavirus Genome Variants Identified in the Territories of Volga Federal District", which reflects the actual content of the article.

Author Response

We would like to thank the reviewers for their very constructive comments and detailed suggestions for the manuscript. We believe that the comments have identified important areas which required improvement. We have revised the text, incorporating all the suggestions made by the two reviewers. Below, you will find a point-by-point description of how each comment was addressed in the manuscript. The changes are in the “Track
Mode” in the manuscript.

Responses to reviewers’ comments: Original reviewer comments in italics, with responses in regular typeface.

REVIEWER 2

The overall assessment of the work is high. The only remark concerns the title of the article. The mention of the Novosibirsk region is incongruously, because studies based on materials from this region are not presented in the article. In our opinion, correct title will be "Analysis of Puumala Orthohantavirus Genome Variants Identified in the Territories of Volga Federal District", which reflects the actual content of the article.

The authors agree with the reviewer’s recommendation and have revised the title to “Analysis of Puumala Orthohantavirus Genome Variants Identified in the Territories of Volga Federal District”

Reviewer 3 Report

The manuscript by Kabwe et al. describes the characterization of Puumala orthohantavirus sequences derived from the reservoir host in three or four sites from the Volga Federal District of Russia. The authors extend similar analyses from (at least) two recent studies in the area (Davidyuk et al. 2020 and Davidyuk et al. 2021) comparing the nucleotide and amino acid similarity of the sequences and making some phylogenetic inferences. The authors attempt to extend this to phylogeographic inferences, but have poorly described these relationships in the manuscript.

The principal criticism I have of the manuscript, is that the authors make inferences using a poorly defined approach. Typically researchers choose either Bayesian or maximum likelihood methods to construct phylogenetic trees, whereas the authors used maximum parsimony methods, so this is rather an issue of conforming to contemporary methods in analyzing hantavirus phylogenetic relationships. They should substantiate why they use maximum parsimony instead of maximum likelihood methods, either with publications or by comparing/demonstrating their similarity here.

They have poorly defined what constitutes “high/low” similarity (line 156 and elsewhere), “high/low” variability (line 236) and even state/imply statistical significance to the comparisons in sequence identities without establishing what “significance” means (line 169).  It is fine to describe these relationships in terms of percent sequence/aa identity, but statistical comparisons were not introduced or used.

The more serious limitation of the study follows from this methodological issue, in that they base some (if not all) of their interpretations on these phylogenies.

First, they include “reference” sequences which are (1) from isolates whose passage number is unknown and (2) from sampling locations that are mostly unknown (or at best generalized). E.g., “DTK-Ufa97” is probably from Ufa in the Bashkiria region 400km to the east of the samples here, and even less is known about the strains “CG1820” and “Kazan”. Compare that to PUUV/Samara_49/CG/2005, whose precise location was published (53.40N 49.60E). Apart from uncertainties in making phylogeographic assessments, it is established that PUUV acquires cell culture adaptations which may interfere with their analysis.

Second, the authors claim to have evidence of reassortment/recombination. Their inferences are based on certain samples “clustering” with one group for one segment, and another group for a different genetic segment. In principal, that would be once piece of evidence for reassortment/recombination, however in every instance, the authors have based their inferences on ancestral nodes with little to no phylogenetic support. In other words, the authors state that they use a cutoff of 70%, but then make inferences and define lineages/sublineages/clades/groups based on nodes without that level of support. Without the support, it is clear that these nodes can “move“ as it indicates there is insufficient phylogenetic information present to resolve that ancestral node. The authors cite Castel et al. 2019, who excellently showed that the DAN lineage “moves” position relative to other lineages (with good support each time!) depending on whether one uses maximum likelihood or Bayesian approaches to construct the tree. In the current study, I would guess that this apparent evidence of reassortment/recombination changes depending on substitution matrix or method used to reconstruct the tree. The authors should remove these portions of the discussion for which they have no support and reformulate their discussion.

Finally, and more of a minor problem, but one that makes it very difficult to understand the study, is the naming conventions. The current study is very interesting in light of the previous findings, but as with the previous studies, they rely on “clusters A/B/C” and trapping groups S1-20 or again poorly defined clades/subclades. Surprisingly, the authors did not cite these previous manuscripts in their discussion about erecting sub-lineages of RUS (paragraph beginning lines 194-197). In this manuscript they simply state “The S segment nucleotide sequences formed three groups corresponding to [the] geographical location of their [sic] bank vole trapping sites” and label them Group A, B, and C. This is rather confusing – particularly when trying to make sense of the current study in light of the previous studies. I had to reconstruct the phylogenetic tree myself, using all previously published samples from this region (unfortunately, the sequences from the current study have not yet been released – but that is to be expected), and compare it to a map of the sites (also constructed myself) where the rodents were reported to be captured. Only then did I understand that this is a very interesting dataset. Simply stated – the manuscript is very poorly organized and confusing. The authors would benefit from re-writing the manuscript, presenting their data in a better way, perhaps with a detailed map and better phylogenetic trees. *Provided* that they perform/justify their analyses according to contemporary standards and base their inferences on solid supporting evidence.

For example, having the “genogroup” or “clade” or whatever listsed in the table of reference sequences would be excellent, as the authors continue to refer to “Pre-Kama” or “Trans-Kama” sequences  in the text.

Here is a running list of some minor and major edits that should be performed. I could not continue past line 176, as I have spent a considerable amount of time reviewing this manuscript as it is. The authors should request a native English speaker, or someone proficient in writing scientific English, review and edit their manuscript.

Line 45:  delete “been”

Line 46: what does “most endemic” mean? An organism is either present or absent…

Line 49: correct article agreement (“has a…form” or “has mild to moderate clinical forms”) and subject-verb agreement “forms, that are often referred to as…” or “form, that is often referred to as”)

Line 51: s-v agreement “the virus…causes a…”

Line 52: “In the previous study”…which previous study? Nothing is cited here.

Lines 52-57: these two sentences seem out of place and completely irrelevant to the study at hand. Furthermore, they don’t report any results from rodents captured in the Novosibirsk region of Siberia, so these sentences can be safely deleted.

Line 61: article agreement “with an infected rodent” or “with infected rodents”

Line 64: “Whether the high cases…” what is meant by “high cases”? High in number? And which cases is this sentence referring to? “the” high cases?

Lines 64 and 65: it seems as if the authors are implying that PUUV incidence in humans is variable throughout the VFD (but have not stated this yet), and they are interested in understanding whether they can associate specific viral genotypes with disease incidence in humans.

Line 75: delete “respectively”

Line 76: “The PUUV genome is highly variable…”

Line 77: “…location of bank bank voles in regions of Europe”

Line 92: “…investigate the genetic characteristics of PUUV in bank voles…”

Line 93-94: what is a “phylogenetic link”? a phylogenetic relationship?

Line 94: delete “Russia”

Line 95-97: “Secondly, we investigated….to obtain first data…”

Lines 97, 139, 141-144: Why include this information about samples from the Novosibirsk region at all? What does this add to the study? If it was a “second aim”, that failed on “technical reasons”, but the technical reasons are not discussed then there is no reason to include them as “negative data”.

Line 109: it was stated that cDNA synthesis was performed with RevertAid RT, and here the authors state that RT-PCR was performed with Taq polymerase… Perhaps they mean “PCR amplification was undertaken using…”

Line 120: “Multiple nucleotide sequence alignments…”

Line 125: How was it decided that T3P was the “optimal substitution model”?

Line 126: “pruning”

Line 125-127: This sentence should be re-written. Maybe it’s 2 sentences?

Line 127-129: “Bootstrap values less than 70% are not shown on the tree.”

Line 128-133: Rewrite this sentence. I suggest “To construct the phylogenies, reference sequences of PUUV strains from multiple lineages were obtained from GenBank, including those previously characterized from the RT and other regions of Russia (Table 1). Sequences of Tula orthohantavirus were used as outgroups for the phylogenetic analysis of the S segment (AF164093), M segment (NC_005228) and L segment (NC_005226).”

Line 137: delete “respectively”

Line 149: “Also”? Maybe the authors could replace this with “In total, 43 bank vole lung tissues…”

Line 149-150: How were 43 tissues “positive for PUUV RNA” but only 11 sequences were obtained? What am I missing? Only RT-PCR was described, so I presume that amplified products (“bands”) were sequenced (from 43 individuals, with 3 bands per individual – S, M, L? = 129 bands?). If only 11 sequences were obtained, then I think that means only 11 rodents were positive for PUUV RNA… Or can the authors provide evidence that these other 32 rodent tissues were PUUV-RNA positive? (i.e., to exclude non-specific amplification?)

Line 150, 156, 177, 179: I suggest defining CDS or just writing “coding-complete sequence”

Line 151: delete the “/”

Line 158: “The S segment…” (delete “all”)

Line 163: “further will be referred to…” where? There is no further reference to “Kurlan-Mullovka”

Line 165: “…was up to 99.8%” I think it was exactly 99.8% because there were only 2 sequences.

Lines 165-166: “Analysis of these sequences among the groups showed a lower identity.” Which sequences? Which groups? In general the use of “among” and “between” seems to be confused on several occasions.

Lines 167-169: “Interestingly, the sequence identities between groups B and C differed significantly although both were from the ULR”. Also, see my note below about “significantly” here – where is the statistical test of significance?

Lines 169-171: “This type of identity…” which type of identity? “…is observed among the PUUV strains…” again ‘among’ is ambiguous here “…which are geographically distantly located from each other”. Please provide the evidence of this – a citation, a statistical comparison, etc.

Lines 173-174: “These identities were lower…” perhaps should be “The sequence identities between all RUS-lineage PUUV strains and all other genetic lineages was 81.3-86.5%.”

Author Response

[Tropicalmed] Manuscript ID: tropicalmed-1583370: Analysis of Puumala Orthohantavirus Genome Variants Identified in the Territories of Volga Federal District.

We would like to thank the reviewers for their very constructive comments and detailed suggestions for the manuscript. We believe that the comments have identified important areas which required improvement. We have revised the text, incorporating all the suggestions made by the two reviewers. Below, you will find a point-by-point description of how each comment was addressed in the manuscript. The changes are in the “Track
Mode” in the manuscript.

Responses to reviewers’ comments: Original reviewer comments in italics, with responses in regular typeface.

REVIEWER 3

The manuscript by Kabwe et al. describes the characterization of Puumala orthohantavirus sequences derived from the reservoir host in three or four sites from the Volga Federal District of Russia. The authors extend similar analyses from (at least) two recent studies in the area (Davidyuk et al. 2020 and Davidyuk et al. 2021) comparing the nucleotide and amino acid similarity of the sequences and making some phylogenetic inferences. The authors attempt to extend this to phylogeographic inferences, but have poorly described these relationships in the manuscript.

The principal criticism I have of the manuscript, is that the authors make inferences using a poorly defined approach. Typically researchers choose either Bayesian or maximum likelihood methods to construct phylogenetic trees, whereas the authors used maximum parsimony methods, so this is rather an issue of conforming to contemporary methods in analyzing hantavirus phylogenetic relationships. They should substantiate why they use maximum parsimony instead of maximum likelihood methods, either with publications or by comparing/demonstrating their similarity here.

The authors agree with the reviewer’s point of view and recommendations. However, the authors would like to state that to analyze the results as in the previous studies we used both MP and ML methods. The infered phylogenetic trees for the S and L segments had almost the same topology and only slightly differed in bootstrap values. Differences were observed in the tree topology for segment M: on the tree constructed using the ML method, the Varaksino strains were branched with Pre-Kama area strains, and on the tree constructed using the MP method, Varaksino strains were closer to Trans-Kama strains. However, bootstraps support in both variants was below 70%. Therefore, we could not make an unambiguous conclusion about the advantage of the ML method in this case. At the same time, the assumption of a possible reassortant origin of the genome of Varaksino strains is based largely on the difference in the topology of the trees of the S and L segments.  As mentioned above, the trees constructed using the MP and ML methods practically do not differ in topology. However, trees constructed using the MP method, in our opinion, are better perceived visually that’s why we used this method to prepare the trees in the manuscript. However, based on the reviewers’ recommendations, we have replaced the MP trees with ML. Also, we have changed the description of the phylogenetic method and made changes to the discussion of the results.

They have poorly defined what constitutes “high/low” similarity (line 156 and elsewhere), “high/low” variability (line 236) and even state/imply statistical significance to the comparisons in sequence identities without establishing what “significance” means (line 169).  It is fine to describe these relationships in terms of percent sequence/aa identity, but statistical comparisons were not introduced or used.

The authors thank the reviewer for identifying these confusing statements and words. The authors interpretation of the data is only based on the percentatiles of nucleotide sequences and aa. The description of our resutls is entirely based on the known percentile differences among PUUV strains within the lineage and intra-lineages. 

The more serious limitation of the study follows from this methodological issue, in that they base some (if not all) of their interpretations on these phylogenies.

First, they include “reference” sequences which are (1) from isolates whose passage number is unknown and (2) from sampling locations that are mostly unknown (or at best generalized). E.g., “DTK-Ufa97” is probably from Ufa in the Bashkiria region 400km to the east of the samples here, and even less is known about the strains “CG1820” and “Kazan”. Compare that to PUUV/Samara_49/CG/2005, whose precise location was published (53.40N 49.60E). Apart from uncertainties in making phylogeographic assessments, it is established that PUUV acquires cell culture adaptations which may interfere with their analysis.

The authors acknowledge the reviewers’ insightful comments and recommendations. However, they would like to state that to conduct the phylogenetic analysis, we required PUUV sequence genome segments of the RUS genetic lineage that would coincide with the genome fragment that we obtained. Also, to have desirable results the S, M, and L segments used for comparisons should come from the same strain. Unfortunately, the Genbank database contains limited number of PUUV strain sequences that meet these conditions and these are from Samara, "Kazan", "CG1820" and "DTK/Ufa-97". The CDS nucleotide sequences of "CG1820" from Bashkiria and "DTK/Ufa-97" strains are 99.8% identity for S segment, 99.6% identity for the M segment, and 99.8% for the L segment. "Kazan" strain was isolated from a bank vole, judging by the name near Kazan, and was introduced into cell culture in 1983 (Gavrilovskaya et al, Archives of Virology 77, 87-90 (1983); Lundkvist et al, Journal of Virological Methods 52 (1995) 75-86). According to CDS S segment nucleotide sequences analysis of PUUV strains from the Republic of Tatarstan, it is most likely that the population of bank voles from which the strain was isolated located in the Pre-Kama area at a distance of 20-40 km east or southeast of the modern eastern border of Kazan. The identity of the nucleotide sequences of Kazan strain and strains from this part of the Pre-Kama area (MW504255, MW504251) exceeds 96%. Therefore, we do not believe that the use of these strains for phylogenetic analysis can seriously affect the interpretation of the results.

Second, the authors claim to have evidence of reassortment/recombination. Their inferences are based on certain samples “clustering” with one group for one segment, and another group for a different genetic segment. In principal, that would be once piece of evidence for reassortment/recombination, however in every instance, the authors have based their inferences on ancestral nodes with little to no phylogenetic support. In other words, the authors state that they use a cutoff of 70%, but then make inferences and define lineages/sublineages/clades/groups based on nodes without that level of support. Without the support, it is clear that these nodes can “move“ as it indicates there is insufficient phylogenetic information present to resolve that ancestral node. The authors cite Castel et al. 2019, who excellently showed that the DAN lineage “moves” position relative to other lineages (with good support each time!) depending on whether one uses maximum likelihood or Bayesian approaches to construct the tree. In the current study, I would guess that this apparent evidence of reassortment/recombination changes depending on substitution matrix or method used to reconstruct the tree. The authors should remove these portions of the discussion for which they have no support and reformulate their discussion.

The authors reconstructed the phylogenetic trees using different models of the ML method. Regardless of the model used on the phylogenetic tree, Varaksino strains were grouped with strains circulating in the Trans-Kama area of the Republic of Tatarstan. Nevertheless, the authors agree that this may not be enough for an unambiguous conclusion about the origin of reassortant genomes of investigated strains. Therefore, the authors reformulated this part of the discussion in the manuscript.

Finally, and more of a minor problem, but one that makes it very difficult to understand the study, is the naming conventions. The current study is very interesting in light of the previous findings, but as with the previous studies, they rely on “clusters A/B/C” and trapping groups S1-20 or again poorly defined clades/subclades. Surprisingly, the authors did not cite these previous manuscripts in their discussion about erecting sub-lineages of RUS (paragraph beginning lines 194-197). In this manuscript they simply state “The S segment nucleotide sequences formed three groups corresponding to [the] geographical location of their [sic] bank vole trapping sites” and label them Group A, B, and C. This is rather confusing – particularly when trying to make sense of the current study in light of the previous studies. I had to reconstruct the phylogenetic tree myself, using all previously published samples from this region (unfortunately, the sequences from the current study have not yet been released – but that is to be expected), and compare it to a map of the sites (also constructed myself) where the rodents were reported to be captured. Only then did I understand that this is a very interesting dataset. Simply stated – the manuscript is very poorly organized and confusing. The authors would benefit from re-writing the manuscript, presenting their data in a better way, perhaps with a detailed map and better phylogenetic trees. *Provided* that they perform/justify their analyses according to contemporary standards and base their inferences on solid supporting evidence.

The authors would like to thank the reviewer for taking time to review the manuscripts offer insightful recommendations and comments. It pleases the authors to hear that the reviewer found the dataset interesting. The authors have reworded and rewritten the manuscript to clearly state the information presented. The authors have gone through all the manuscript to improve on the results presentation.

For example, having the “genogroup” or “clade” or whatever listed in the table of reference sequences would be excellent, as the authors continue to refer to “Pre-Kama” or “Trans-Kama” sequences in the text.

The authors made changes to the names of strain groups and subclades in the text and Figures 2, 3 and 4, and also added information about the localization of reference PUUV strains in the regions of the Republic of Tatarstan to Table 1. Corresponding changes were also made to the results and discussion.

Here is a running list of some minor and major edits that should be performed. I could not continue past line 176, as I have spent a considerable amount of time reviewing this manuscript as it is. The authors should request a native English speaker, or someone proficient in writing scientific English, review and edit their manuscript.

The authors thank the reviewer for his insight recommendations and comment for identifying this part which needs improvement. Sections throughout have been edited and changed including the remaing sections that the reviewer couldn’t manage to review. The entire manuscript has been checked by a native English Speaker.

Line 45:  delete “been”

The authors agree and the word “been” has been deleted from the sentence.

Line 46: what does “most endemic” mean? An organism is either present or absent…

The sentence has been updated to “the most affected regions with HFRS”

Line 49: correct article agreement (“has a…form” or “has mild to moderate clinical forms”) and subject-verb agreement “forms, that are often referred to as…” or “form, that is often referred to as”)

The authors have corrected the article agreement

Line 51: s-v agreement “the virus…causes a…”

The authors apologize for the typographical mistakes. The verb has been corrected.

Line 52: “In the previous study”…which previous study? Nothing is cited here.

The authors removed this fragment from the manuscript.

Lines 52-57: these two sentences seem out of place and completely irrelevant to the study at hand. Furthermore, they don’t report any results from rodents captured in the Novosibirsk region of Siberia, so these sentences can be safely deleted.

The authors wish to thank the reviewer for identifying the limited results about Novosibirsk. The sentence has been removed.

Line 61: article agreement “with an infected rodent” or “with infected rodents”

The authors apologize for the typographical mistakes. The article has been corrected.

Line 64: “Whether the high cases…” what is meant by “high cases”? High in number? And which cases is this sentence referring to? “the” high cases?

The sentence has been corrected

Lines 64 and 65: it seems as if the authors are implying that PUUV incidence in humans is variable throughout the VFD (but have not stated this yet), and they are interested in understanding whether they can associate specific viral genotypes with disease incidence in humans.

The sentence has been reworded.

Line 75: delete “respectively”

The authors agree and the word “respectively” has been deleted

Line 76: “The PUUV genome is highly variable…”

The authors have made the typographical change

Line 77: “…location of bank voles in regions of Europe”

The authors have made the typographical change

Line 92: “…investigate the genetic characteristics of PUUV in bank voles…”

The authors agree and the typographical change

Line 93-94: what is a “phylogenetic link”? a phylogenetic relationship?

The authors have deleted the word “link” and replaced it with “relationship”

Line 94: delete “Russia”

The authors agree and have made the typographical change

Line 95-97: “Secondly, we investigated….to obtain first data…”

The entire sentence has been deleted and the goals have been corrected.

Lines 97, 139, 141-144: Why include this information about samples from the Novosibirsk region at all? What does this add to the study? If it was a “second aim”, that failed on “technical reasons”, but the technical reasons are not discussed then there is no reason to include them as “negative data”.

The authors agree with the reviewer’s point of view and would like to thank him for identifying this part of the results which is limited. The information about Novosibirsk region samples has been removed.

Line 109: it was stated that cDNA synthesis was performed with RevertAid RT, and here the authors state that RT-PCR was performed with Taq polymerase… Perhaps they mean “PCR amplification was undertaken using…”

The authors agree and have made the typographical change

Line 120: “Multiple nucleotide sequence alignments…”

The authors have made the typographical change

Line 125: How was it decided that T3P was the “optimal substitution model”?

This sentence has been removed. ML method has been used to infer the phylogenies.

Line 126: “pruning”

The authors agree and have made the typographical change

Line 125-127: This sentence should be re-written. Maybe it’s 2 sentences?

The authors have reworded the sentence

Line 127-129: “Bootstrap values less than 70% are not shown on the tree.”

The authors have made changes in line with reviewer’s recommendations

Line 128-133: Rewrite this sentence. I suggest “To construct the phylogenies, reference sequences of PUUV strains from multiple lineages were obtained from GenBank, including those previously characterized from the RT and other regions of Russia (Table 1). Sequences of Tula orthohantavirus were used as outgroups for the phylogenetic analysis of the S segment (AF164093), M segment (NC_005228) and L segment (NC_005226).”

We thank the reviewer for spotting this oversight and the sentences have been corrected in line with reviewer’s recommendations

Line 137: delete “respectively”

We have made the typographical change

Line 149: “Also”? Maybe the authors could replace this with “In total, 43 bank vole lung tissues…”

The authors have made the correction

Line 149-150: How were 43 tissues “positive for PUUV RNA” but only 11 sequences were obtained? What am I missing? Only RT-PCR was described, so I presume that amplified products (“bands”) were sequenced (from 43 individuals, with 3 bands per individual – S, M, L? = 129 bands?). If only 11 sequences were obtained, then I think that means only 11 rodents were positive for PUUV RNA… Or can the authors provide evidence that these other 32 rodent tissues were PUUV-RNA positive? (i.e., to exclude non-specific amplification?)

The authors agree that the sentence is somewhat confusing; therefore, the sentence was changed to reflect and clarify the meaning. Also, added information about initial screening of samples to Materials and Methods.

Line 150, 156, 177, 179: I suggest defining CDS or just writing “coding-complete sequence”

The authors agree with the reviewer and have defined CDS in the text

Line 151: delete the “/”

The authors agree and have the typographical change

Line 158: “The S segment…” (delete “all”)

We have deleted the word “all”

Line 163: “further will be referred to…” where? There is no further reference to “Kurlan-Mullovka”

We thank the reviewer for spotting this oversight and the term “Kurlan-Mullovka” was used in manuscript.

Line 165: “…was up to 99.8%” I think it was exactly 99.8% because there were only 2 sequences.

“Up to” has been deleted

Lines 165-166: “Analysis of these sequences among the groups showed a lower identity.” Which sequences? Which groups? In general, the use of “among” and “between” seems to be confused on several occasions.

The authors apologize for any confusion. The sentence has been edited and words “among” and “between” checked.

Lines 167-169: “Interestingly, the sequence identities between groups B and C differed significantly although both were from the ULR”. Also, see my note below about “significantly” here – where is the statistical test of significance?

The authors thank the reviewer for identifying this oversight and misleading use of significant. The word significantly has therefore been deleted and the sentence has been reworded.

Lines 169-171: “This type of identity…” which type of identity? “…is observed among the PUUV strains…” again ‘among’ is ambiguous here “…which are geographically distantly located from each other”. Please provide the evidence of this – a citation, a statistical comparison, etc.

This sentence has been reworded and extra information added to improve the meaning

Lines 173-174: “These identities were lower…” perhaps should be “The sequence identities between all RUS-lineage PUUV strains and all other genetic lineages was 81.3-86.5%.”

We thank the reviewer for this insight recommendation. The sentence has been reworded in line with the reviewers’ recommendations.

Round 2

Reviewer 1 Report

Thank you for taking into account my suggestion.

Reviewer 3 Report

The authors have greatly improved their manuscript. They have been more precise with their description of methods and presentation of results. Including region names instead of A/B/C made reading this much much more clear.

As I mentioned in a previous review, I think the authors have an interesting dataset. They could make a nice contribution to the study of PUUV - a field which has many many fine manuscripts already that describe the genetic relatedness of strains and further make phylogenetic and phylogeographic inferences about the relationships between lineages/sublineages. The latter aspect is particularly important, as the field has relatively few studies which characterize genetic diversity on larger geographic scales - and particularly very few from the RUS lineage! Most rather focus on the evolution of PUUV over small distances/time scales, and/or describe spatial patterns of genetic diversity within small geographic areas.

This manuscript could contribute a phylogeographic analysis of PUUV in an under-studied region & within an understudied lineage. It is a shame that the authors did not do this, as the authors have already described many strains from the surrounding area - linking the regions described herein -  using the same techniques! I was a bit disappointed to read that the very interesting finding about how this relates to bank vole migration and PUUV evolution was discussed in a few sentences at the end. This is indeed a major area of research in PUUV. Instead the authors focus on reassortment/recombination of the Veraksino strains. This is an area of research which is rather debated in PUUV - perhaps thought to occur sporadically. Therefore it is a finding that carries a large burden of support, and a finding that the authors have very weak support for. Most of the reason for the weak support is in the lack of rigor in their phylogenetic analysis.

The phylogenetic analysis again makes inferences based on ancestral relationships (nodes) with low support.

My suggestion for the authors is to review the current literature and consider some questions: What is the relationship between geographic distance and phylogenetic distance for PUUV? I think their dataset can provide this analysis - and they hint at this several times (e.g., line 161 there is and "interesting" level of sequence identity between K-M and Barysh..."one that is simlar "among the PUUV strains, which are geographically distantly located from each other."

However they do not pursue this idea in the manuscript. They rather focus on the clustering of Veraksino with other strains. While I agree that the S cds shares a common ancestor with Pre-Kama genotypes (Fig2, 74%) and the partial-L segment shares a common ancestor with the Trans-Kama genotype (Fig 4, 71%) , there is no support for a common ancestral relationship between partial M segment and either Trans- or Pre-Kama genotypes.

Moreover, I disagree (and rather fail to see the point) with classifying various heirarchies of relatedness for each phylogeny - particularly when the group is not supported at the stated bootstrap level. Is "sublineage" or geographic clade a meaningful unit here? Could the authors make a data-supported scientific argument as to why these are biological groupings? (e.g., see above with my leading question about phylogeographic relationships among PUUV strains). I agree that Barysh strains are a distinctive genotype from the remaining RUS lineage - they are highly supported in each figure.

The authors did not respond adequately to my previous request to justify the substitution method used to generate their phylogenies (TN93), as is common contemporary practice. Why/how was this model chosen?

There is no scale on the map.

In sum, this manuscript describes the genetic diversity - descriptive statistics of sequence identity - of PUUV in an under-studied region. The information is good, and will be welcomed by others in the field that are interested in hantavirus phylogeography and evolution. The descriptions of sequence identity (Tables 1 and 2) provide some scientific value, although it is something that one could also quickly understand by aligning publicly available sequences. The manuscript provides relatively specific trapping locations that will be helpful in future analyses of phylogeography. The authors would improve the impact of their manuscript by conforming to contemporary approaches in testing virus phylogenetics. As it stands there are numerous problems with their phylogenetic approach, which I have now mentioned in both reviews of this manuscript.